# Magnetic and electronic phase transitions probed by nanomechanical resonators

Makars Šiškins [1,4✉], Martin Lee [1,4], Samuel Mañas-Valero [2], Eugenio Coronado [2], Yaroslav M. Blanter[1], Herre S. J. van der Zant [1✉] & Peter G. Steeneken [1,3✉]

The reduced dimensionality of two-dimensional (2D) materials results in characteristic types of magnetically and electronically ordered phases. However, only few methods are available to study this order, in particular in ultrathin insulating antiferromagnets that couple weakly to magnetic and electronic probes. Here, we demonstrate that phase transitions in thin membranes of 2D antiferromagnetic $FePS_3$, $MnPS_3$ and $NiPS_3$ can be probed mechanically via the temperature-dependent resonance frequency and quality factor. The observed relation between mechanical motion and antiferromagnetic order is shown to be mediated by the specific heat and reveals a strong dependence of the Néel temperature of $FePS_3$ on electrostatically induced strain. The methodology is not restricted to magnetic order, as we demonstrate by probing an electronic charge-density-wave phase in $2H\text{-}TaS_2$. It thus offers the potential to characterize phase transitions in a wide variety of materials, including those that are antiferromagnetic, insulating or so thin that conventional bulk characterization methods become unsuitable.

[1] Kavli Institute of Nanoscience, Delft University of Technology, Lorentzweg 1, 2628 CJ Delft, The Netherlands. [2] Instituto de Ciencia Molecular (ICMol), Universitat de València, c/Catedrático José Beltrán 2, 46980 Paterna, Spain. [3] Department of Precision and Microsystems Engineering, Delft University of Technology, Mekelweg 2, 2628 CD Delft, The Netherlands. [4] These authors contributed equally: Makars Šiškins, Martin Lee. ✉email: m.siskins-1@tudelft.nl; h.s.j.vanderzant@tudelft.nl; p.g.steeneken@tudelft.nl

Nanomechanical resonators made of two-dimensional (2D) materials offer interesting pathways for realizing high-performance devices[1,2]. Unique functionalities and phenomena emerge when combining nanomechanics with the types of magnetic and electronic phases that have recently been uncovered in 2D materials like magic-angle induced phase transitions[3,4], 2D Ising antiferromagnets[5] and ferromagnetism in 2D atomic layers[6,7] and heterostructures[8]. Only a few methods are available to study these phases in 2D materials[5–9]. A universal method to characterize phase transitions in bulk crystals is via anomalies in the specific heat, that are present at the transition temperature according to Landau's theory[10]. However, specific heat is difficult to measure in thin micron-sized samples with a mass of less than a picogram[11,12].

We demonstrate that these phases are strongly coupled to mechanical motion: the temperature-dependent resonance frequency and quality factor of multilayer 2D material membranes show anomalies near the phase transition temperature. Although coupling between mechanical and electronic/magnetic degrees of freedom might not seem obvious, the intuitive picture behind this coupling is that changes in the electronic/magnetic order and entropy in a material are reflected in its specific heat, which in turn results in variations in the thermal expansion coefficient that affect the tension and resonance frequency. As the specific heat near a phase transition is expected to exhibit a discontinuity[10], the temperature-dependent resonance frequency of a suspended membrane can thus be used to probe this transition.

The coupling of mechanical degrees of freedom to magnetic and electronic order is attributed to thermodynamic relations.

Nanomechanical resonators, therefore, offer the potential to characterize phase transitions and realize device concepts in a wide variety of systems, not restricted only to van der Waals materials but including those that are ultrathin, antiferromagnetic or insulating[8]. Here, we use nanomechanical motion to investigate magnetic order in membranes of semiconducting $FePS_3$, $NiPS_3$ and insulating $MnPS_3$—antiferromagnetic members of the transition-metal phosphor trisulphides $(MPS_3)$[13], and subsequently discuss results on metallic $2H$-$TaS_2$, which exhibits a transition to a charge density wave state[14].

## Results

**Antiferromagnetic mechanical resonators.** $FePS_3$ is an Ising-type antiferromagnet with a Néel temperature in bulk in the range of $T_N \sim 118$–$123\,K$[5,13,15], exhibiting a distinct feature in its specific heat near $T_N$[15]. Ionic layers in $FePS_3$ are stacked in van der Waals planes, that can be exfoliated to thin the crystal down with atomic precision[5]. Using mechanical exfoliation and all-dry viscoelastic stamping[16], we transfer thin flakes of $FePS_3$ over circular cavities etched in an oxidised Si wafer, to form membranes (see the inset in Fig. 1a). Suspended $FePS_3$ devices with thicknesses ranging from 8 to 45 nm are placed in a cryostat and cooled down to a temperature of 4 K. The resonance frequency of the nanodrums is then characterized using a laser interferometry technique[17] (see Fig. 1a and "Methods").

The resonance frequency of the fundamental membrane mode, $f_0(T)$, is measured in the temperature range from 4 to 200 K. Typical resonances are shown in Fig. 1b–d in the

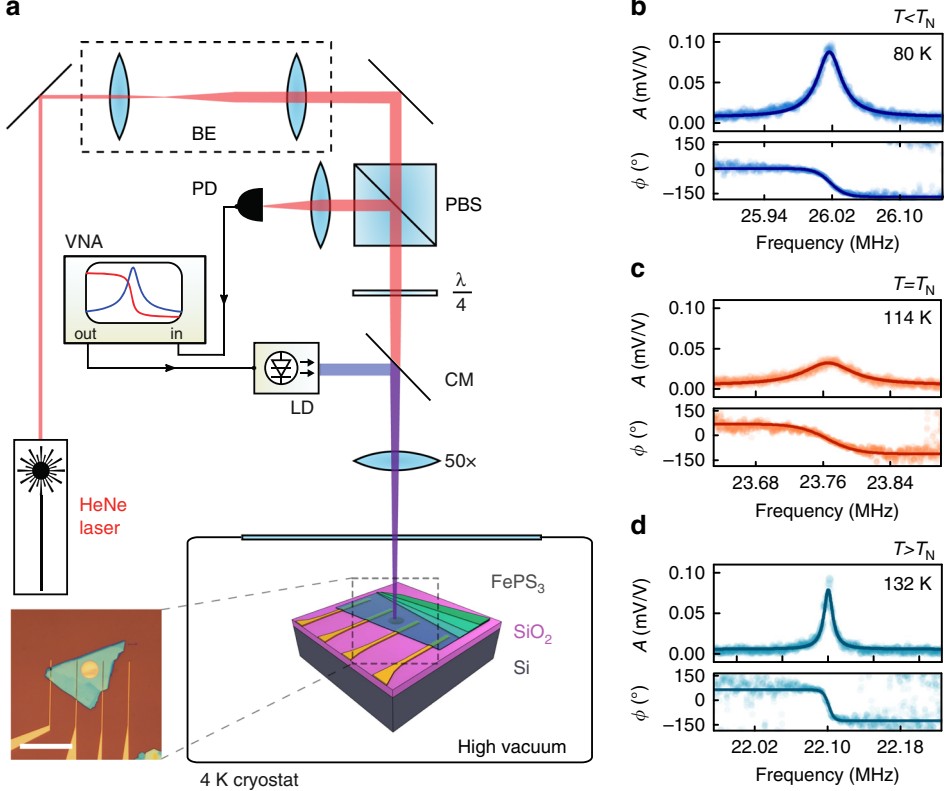

**Fig. 1 Characterisation of mechanical resonances in a thin antiferromagnetic FePS₃ membrane. a** Laser interferometry setup. Red interferometric detection laser: $\lambda_{red} = 632\,nm$. Blue actuation laser diode: $\lambda_{blue} = 405\,nm$. VNA, vector network analyzer, CM, cold mirror; PBS, polarizing beam splitter; PD, photodiode; LD, laser diode. Inset: optical image of a FePS₃ membrane, including electrodes introducing an option for electrostatic control of strain in the membrane. Flake thickness: $45.2 \pm 0.6\,nm$; membrane diameter: $d = 10\,\mu m$. Scale bar: $30\,\mu m$. **b–d** Amplitude ($A$) and phase ($\phi$) of the fundamental resonance at three different temperatures for the device shown in (**a**). Filled dots, measured data; solid lines, fit of the mechanical resonance used to determine $f_0$ and $Q$[17].

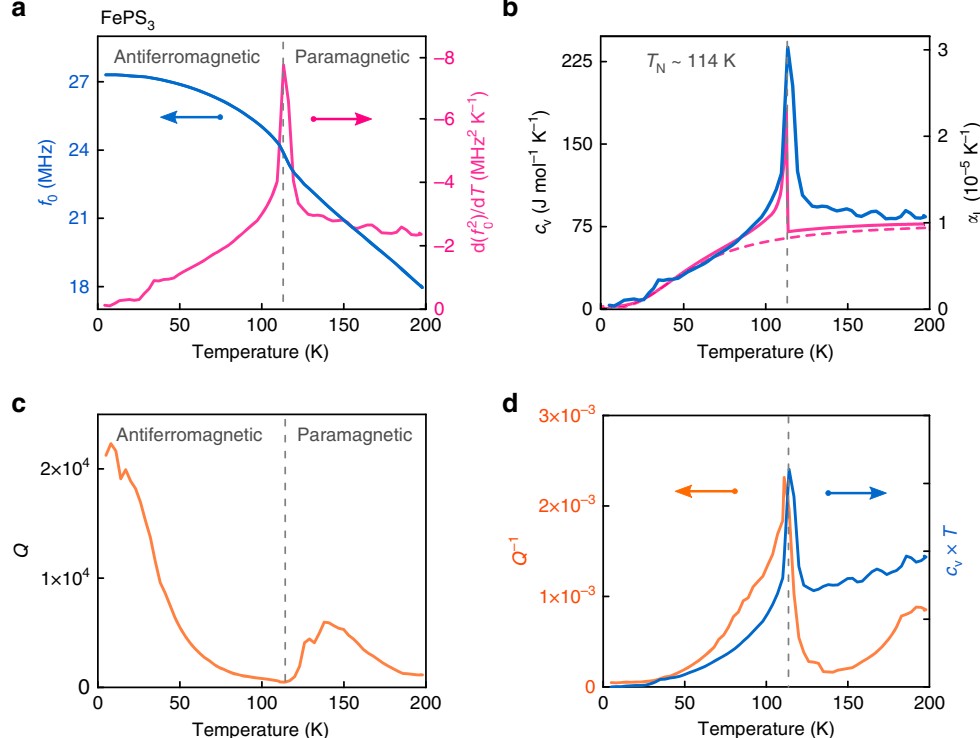

**Fig. 2 Mechanical and thermal properties of a FePS$_3$ resonator with membrane thickness of 45.2 ± 0.6 nm.** In all panels, dashed vertical lines indicate the detected transition temperature, $T_N$ = 114 ± 3 K as determined from the peak in the temperature derivative of $f_0^2$. **a** Solid blue line—measured resonance frequency as a function of temperature. Solid magenta line—temperature derivative of $f_0^2$. **b** Solid blue line—experimentally derived specific heat and corresponding thermal expansion coefficient. Solid magenta line—the theoretical calculation of the magnetic specific heat as reported in Takano et al.[15] added to the phononic specific heat from Debye model (dashed magenta line) with a Debye temperature of $\Theta_D$ = 236 K[15]. **c** Mechanical quality factor $Q$ $(T)$ of the membrane fundamental resonance. **d** Solid orange line—measured mechanical damping $Q^{-1}(T)$ as a function temperature. Solid blue line— normalized $c_v(T)\,T$ term[20,21] (see Supplementary equation (14)), with $c_v(T)$ taken from (**b**).

antiferromagnetic phase (80 K), near the transition (114 K) and in the paramagnetic phase (132 K), respectively. Figure 2a shows $f_0(T)$ of the same FePS$_3$ membrane (solid blue curve). Near the phase transition, significant changes in amplitude, resonance frequency, and quality factor are observed.

**Resonance and specific heat**. To analyze the data further, we first analyze the relation between $f_0$ and the specific heat. The decrease in resonance frequency with increasing temperature in Fig. 2a is indicative of a reduction in strain due to thermal expansion of the membrane. The observed changes can be understood by considering the resonance frequency of a bi-axially tensile strained circular membrane:

$$f_0(T) = \frac{2.4048}{\pi d}\sqrt{\frac{E}{\rho}\frac{\epsilon(T)}{(1-\nu)}}, \quad (1)$$

where $E$ is the Young's modulus of the material, $\nu$ its Poisson's ratio, $\rho$ its mass density, $\epsilon(T)$ the strain and $T$ the temperature. The linear thermal expansion coefficient of the membrane, $\alpha_L(T)$, and silicon substrate, $\alpha_{Si}(T)$, are related to the strain in the membrane[18] as $\frac{d\epsilon(T)}{dT} \approx -(\alpha_L(T) - \alpha_{Si}(T))$, using the approximation $\alpha_{SiO_2} \ll \alpha_{Si}$ (see Supplementary Note 1). By combining the given expression for $\frac{d\epsilon(T)}{dT}$ with equation (1) and by using the thermodynamic relation $\alpha_L(T) = \gamma c_v(T)/(3KV_M)$[19] between $\alpha_L(T)$ and the specific heat (molar heat capacity) at constant volume, $c_v(T)$, we obtain:

$$c_v(T) = 3\alpha_L(T)\frac{KV_M}{\gamma} = 3\left(\alpha_{Si} - \frac{1}{\mu^2}\frac{d[f_0^2(T)]}{dT}\right)\frac{KV_M}{\gamma}. \quad (2)$$

Here, $K$ is the bulk modulus, $\gamma$ the Grüneisen parameter, $V_M = M/\rho$ the molar volume of the membrane and $\mu = \frac{2.4048}{\pi d}\sqrt{\frac{E}{\rho(1-\nu)}}$, that are assumed to be only weakly temperature dependent. The small effect of non-constant volume ($\nu \neq 0.5$) on $c_v$ is neglected.

We use the equation (2) to analyze $f_0(T)$ and compare it to the calculated specific heat for FePS$_3$ from literature[15]. In doing so, we estimate the Grüneisen parameter following the Belomest-nykh − Tesleva relation $\gamma \approx \frac{3}{2}\left(\frac{1+\nu}{2-3\nu}\right)$[19,22]. This is an approximation to Leont'ev's formula[23], which is a good estimation of $\gamma$ for bulk isotropic crystalline solids within ~10% of uncertainty[19]. Furthermore, we use literature values for the elastic parameters of FePS$_3$ as obtained from first-principles theoretical calculations[24] to derive $E$ = 103 GPa, $\nu$ = 0.304 and $\rho$ = 3375 kg m$^{-3}$ (see Supplementary Note 2).

**Detecting phase transitions**. In Fig. 2a, the steepest part of the negative slope of $f_0(T)$ (solid blue curve) leads to a large peak in $\frac{d(f_0^2(T))}{dT}$ (solid magenta curve) near 114 K, the temperature which we define as $T_N$ and indicate by the vertical dashed lines. In Fig. 2b the specific heat curve of FePS$_3$ (blue solid line) as estimated from the data in Fig. 2a and equation (2) is displayed. The results are compared to a theoretical model for the specific heat of FePS$_3$ (magenta solid line in Fig. 2b), which is the sum of a phononic contribution from the Debye model (magenta dashed line) and a magnetic contribution as calculated by Takano et al.[15]. It is noted that other, e.g. electronic contributions to $c_v(T)$ are small and can be neglected in this comparison, as is supported by experiments on the specific heat in bulk FePS$_3$ crystals[15]. The

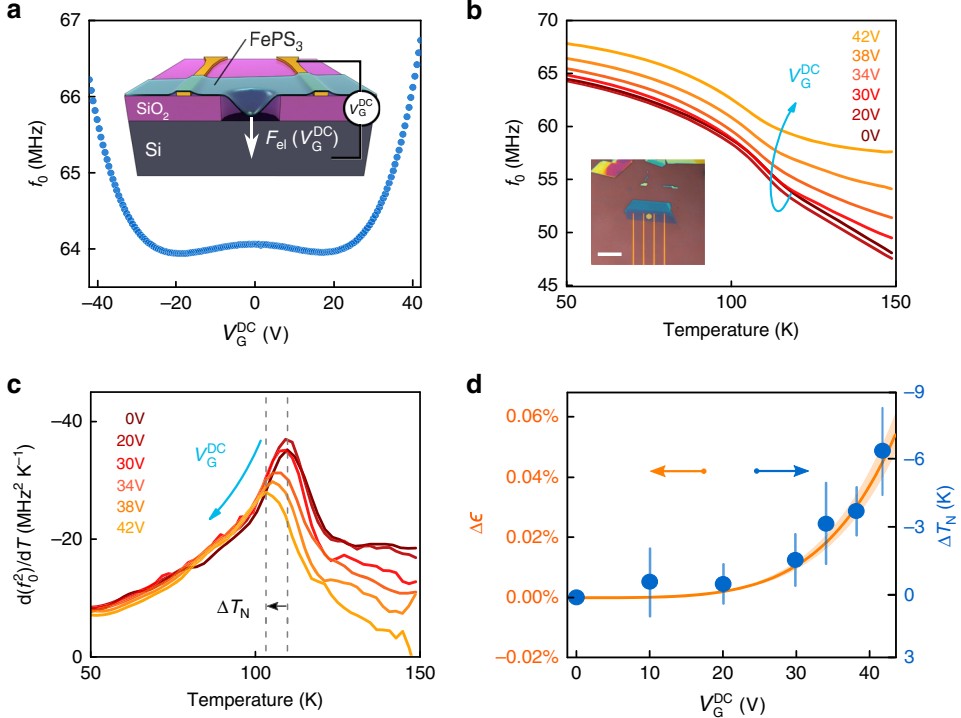

**Fig. 3 Resonance frequency and transition temperature tuning with a gate voltage. a** Resonance frequency as a function of gate voltage at 50 K. Inset: schematics of the electrostatic tuning principle. **b** Resonance frequency as a function of temperature for six different voltages. Inset: optical image of the sample, $t = 8 \pm 0.5$ nm. Scale bar: 16 μm. **c** Derivative of $f_0^2$ as a function of gate voltage and temperature. Blue arrow, line colors and legend indicate the values of $V_G^{DC}$. Dashed gray lines indicate the decrease in transition temperature $\Delta T_N = T_N^*(V_G^{DC}) - T_N(0\,\text{V})$ with increasing $V_G^{DC}$. **d** Blue solid dots—shift in $T_N$ as a function of $V_G^{DC}$ extracted from the peak position in (**c**). Vertical blue bars—error bar in $\Delta T_N$ estimated from determining the peak position in (**c**) within 2% accuracy in the measured maximum. Orange solid line—model of electrostatically induced strain $\Delta \epsilon$ as a function of $V_G^{DC}$ (see Supplementary Note 5).

close correspondence in Fig. 2b between the experimental and theoretical data for $c_v(T)$ supports the applicability of equation (2). It also indicates that changes in the Young's modulus near the phase transition, that can be of the order of a couple of percent[25,26], are insignificant and that it is the anomaly in $c_v$ of FePS$_3$ which produces the observed changes in resonance frequency and the large peak in $\frac{d(f_0^2)}{dT}$ visible in Fig. 2a.

**Effect of strain**. The abrupt change in $c_v(T)$ of the membrane can be understood from Landau's theory of phase transitions[10]. To illustrate this, we consider a simplified model for an anti-ferromagnetic system, like FePS$_3$, with free energy, $F$, which includes a strain-dependent magnetostriction contribution (see Supplementary Note 3). Near the transition temperature and in the absence of a magnetic field it holds that:

$$F = F_0 + [a(T - T_N) + \zeta(\epsilon)]L_z^2 + BL_z^4. \quad (3)$$

Here, $a$ and $B$ are phenomenological positive constants, $L_z$ is the order parameter in the out-of-plane direction and $\zeta(\epsilon) = \eta_{ij}\epsilon_{ij}$, a strain-dependent parameter with $\eta_{ij}$ a material-dependent tensor, that includes the strain and distance-dependent magnetic exchange interactions between neighboring magnetic moments. By minimizing equation (3) with respect to $L_z$, the equilibrium free energy, $F_{min}$, and order parameter are obtained (see Supplementary Note 3). Two important observations can be made. Firstly, strain shifts the transition temperature according to:

$$T_N^*(\epsilon) = T_N - \frac{\zeta(\epsilon)}{a}, \quad (4)$$

where $T_N^*$ is the Néel temperature, below which free energy

minima $F_{min}$ with finite order ($L_z \neq 0$) appear. Secondly, since close to the transition the specific heat follows $c_v(T) = -T\frac{\partial^2 F_{min}}{\partial T^2}$, this general model predicts a discontinuity in $c_v$ of magnitude $T_N^*\frac{a^2}{2B}$ at the transition temperature $T_N^*$, in accordance with the experimental jump in $c_v(T)$ and $\frac{d(f_0^2(T))}{dT}$ observed in Fig. 2a and b.

**Temperature-dependent Q-factor**. We now analyze the quality factor data shown in Fig. 2c, d. Just above $T_N$, the quality factor of the resonance (Fig. 2c) shows a significant increase as the temperature is increased from 114 to 140 K. The observed minimum in the quality factor near the phase transition, suggests that dissipation in the material is linked to the thermodynamics and can be related to thermoelastic damping. We model the thermoelastic damping according to Zener[20] and Lifshitz-Roukes[21] that report dissipation of the form $Q^{-1} = \beta c_v(T)\, T$, where $\beta$ is the thermomechanical term (see Supplementary Note 4). Since we have obtained an estimate of $c_v(T)$ from the resonance frequency analysis (Fig. 2b), we use this relation to compare the experimental dissipation $Q^{-1}(T)$ (orange solid line) to a curve proportional to $c_v(T)\, T$ (blue solid line) in Fig. 2d. Both the measured dissipation and the thermoelastic term display a peak near $T_N \sim$ 114 K. The close qualitative correspondence between the two quantities is an indication that the thermoelastic damping related term indeed can account for the temperature dependence of $Q(T)$ near the phase transition. We note that the temperature-dependent dissipation in thin membranes is still not well understood, and that more intricate effects might play a role in the observed temperature dependence.

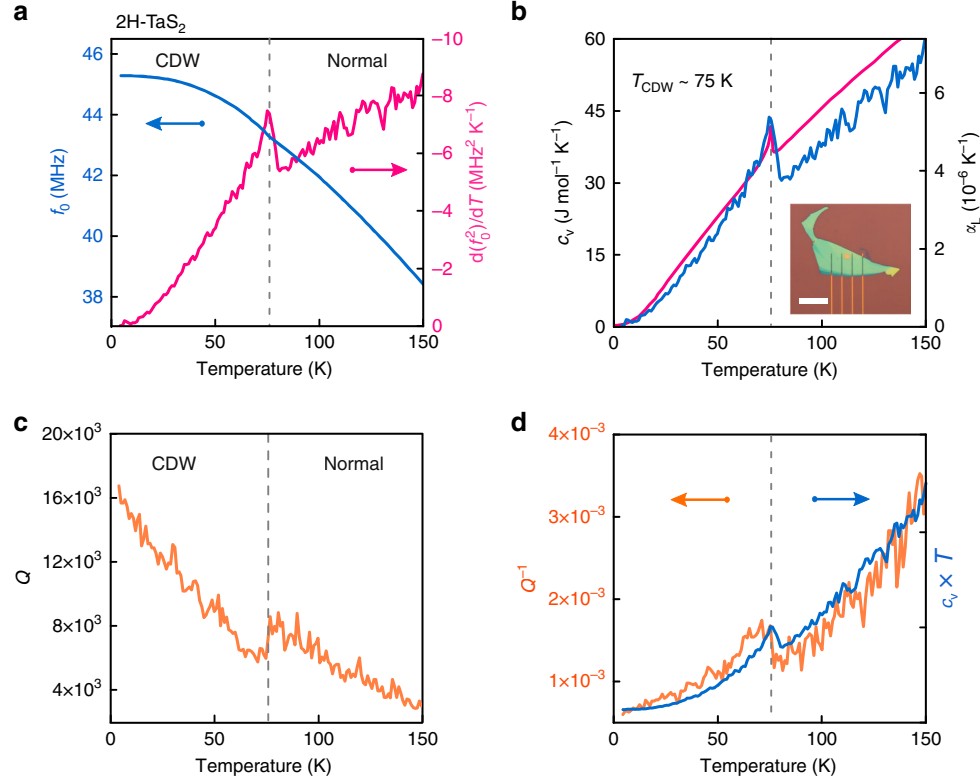

**Fig. 4 Mechanical properties of a 2H-TaS$_2$ resonator with membrane thickness of 31.2 ± 0.6 nm and $d$ = 4 µm.** Dashed vertical line in all 4 panels indicates the detected $T_{CDW}$, that is defined by the peak in $\frac{d(f_0^2(T))}{dT}$. **a** Solid blue line—resonance frequency as a function of temperature. Solid magenta line—temperature derivative of $f_0^2$. **b** Solid blue line—experimentally derived $c_v$ and thermal expansion coefficient as a function of temperature (see Supplementary Note 6). Solid magenta line—specific heat of bulk 2H-TaS$_2$ as reported in Abdel-Hafiez et al.[14]. Inset: optical image of the sample. Scale bar: 20 µm. **c** Quality factor $Q(T)$ as a function of temperature. **d** Solid orange line—measured mechanical damping $Q^{-1}(T)$ as a function of temperature. Solid blue line—curve proportional to the term $c_v(T)\,T$[20,21] (see Supplementary equation (14)), with $c_v(T)$ taken from the experimental data in (**b**).

**Electrostatic strain.** Equation (4) predicts that the transition temperature is strain-dependent due to the distance-dependent interaction coefficient $\zeta(\epsilon)$ between magnetic moments. To verify this effect, we use an 8 ± 0.5 nm thin sample of FePS$_3$ suspended over a cavity of 4 µm in diameter. A gate voltage $V_G^{DC}$ is applied between the flake and the doped bottom Si substrate to introduce an electrostatic force that pulls the membrane down and thus strains it (see Supplementary Figs. 4 and 5). As shown in Fig. 3a, the resonance frequency of the membrane follows a W-shaped curve as a function of gate voltage. This is due to two counteracting effects[27]: at small gate voltages capacitive softening of the membrane occurs, while at higher voltages the membrane tension increases due to the applied electrostatic force, which causes the resonance frequency to increase.

Figure 3b shows $f_0(T)$ for six different gate voltages. The shift of the point of steepest slope of $f_0(T)$ with increasing $V_G^{DC}$ is well visible in Fig. 3b and even more clear in Fig. 3c, where the peak in $\frac{d(f_0^2)}{dT}$ shifts 6 K downward by electrostatic force induced strain. The observed reduction in $T_N^*$ as determined by the peak position in $\frac{d(f_0^2)}{dT}$ qualitatively agrees with the presented model and its strain dependence from equation (4), as shown in Fig. 3d indicative of a reduced coupling of magnetic moments with increasing distance between them due to tensile strain.

## Discussion

Since the coupling between specific heat and the order parameter in materials is of a general thermodynamic nature, the presented methodology is applicable to a wide variety of materials provided

that elastic properties of the material and Grüneisen parameter are weakly temperature dependent, the substrate satisfies the condition $\alpha_{substrate} \ll \alpha_{material}$ and that the frequency shifts and changes in $Q$ are large enough to be resolved. We further demonstrate the method by detecting magnetic phase transitions in NiPS$_3$ and MnPS$_3$. Compared to FePS$_3$, the effect of the phase transitions in MnPS$_3$ and NiPS$_3$ on the resonances is more gradual (see Supplementary Fig. 2) with both materials showing broader maxima in $\frac{d(f_0^2(T))}{dT}$ near their $T_N$ at 76 K and 151 K, respectively, which is consistent with measurements of bulk crystals[13,15].

In order to demonstrate the detection of an electronic phase transition, we now discuss results for 2H-TaS$_2$ that in bulk exhibits a charge density wave (CDW) transition at $T_{CDW}$ ~ 77 K[14]. Figure 4a shows a transition-related anomaly in both $f_0(T)$ (solid blue line) and the temperature derivative of $f_0^2(T)$ (solid magenta line) that peaks at 75 ± 3 K. We convert $\frac{d(f_0^2(T))}{dT}$ to the corresponding $c_v(T)$ using the same approach as discussed before (see Supplementary Note 6). Figure 4b shows a downward step in the specific heat at 75 K (solid blue line), indicative of a phase transition from the CDW to the disordered high-temperature state[10,28] with a close quantitative correspondence to $c_v$ measured in a bulk crystal[14] (drawn magenta line). This anomaly occurs near the electrically determined phase transition temperature of ~77 K on the same flake (see Supplementary Fig. 6c) and is also consistent with the CDW transition temperature previously reported in 2H-TaS$_2$[14]. The Q-factor also shows a local minimum with a drop next to the transition temperature (see Fig. 4c). As discussed before[20,21], $Q^{-1}(T)$ is expected to follow the

same trend as $c_v(T) \, T$. Both quantities are displayed in Fig. 4d and indeed show a good qualitative correspondence.

In conclusion, we have demonstrated that there exist a strong coupling between mechanical motion and order in ultrathin membranes of 2D materials. An analytical equation for the relation between the specific heat of the material and the temperature-dependent resonance frequency is derived and shown to be in good agreement with experimental results. Since the materials are utilized in a suspended state, substrate effects on the electronic and magnetic properties of the thin materials are excluded. The technique is not only appealing for the characterisation of ultra-thin membranes of antiferromagnetic and insulating materials that are difficult to characterize otherwise, but also for the development of device concepts exploiting the unique properties of the materials involved. It is anticipated that it can be applied to a large range of van der Waals materials[8,9], 2D ferromagnets[29], thin 2D complex oxide sheets[30,31] and organic antiferromagnets[32].

## Methods

**Sample fabrication.** To realize electrical contact to the samples for electrostatic experiments, Ti/Au electrodes are pre-patterned by a lift-off technique. Cavities are defined by reactive ion etching of circular holes with a diameter of 4–10$\mu$m in oxidized doped silicon wafers with an $SiO_2$ thickness of 285 nm. Flakes of van der Waals crystals are exfoliated from high quality synthetically grown crystals with known stoichiometry (see Supplementary Note 7). All flakes are transferred on a pre-patterned chip by an all-dry viscoelastic stamping directly after exfoliation. Subsequently, samples are kept in an oxygen-free environment to avoid degradation. In total, data on measurements of three $FePS_3$, one 2H-$TaS_2$, one $NiPS_3$ and one $MnPS_3$ devices is presented in this manuscript.

**Controlled measurement environment.** The samples are mounted on a piezo-based $xy$ nanopositioning stage inside a chamber of a closed-cycle cryostat with optical access. A closed feedback loop controlled local sample heater is used to perform temperature sweeps at a rate of ~5 K per min, while keeping the pressure in the chamber below $10^{-6}$ mbar. During the data acquisition temperature is kept constant with ~10 mK stability.

**Laser interferometry.** A blue diode laser ($\lambda$ = 405 nm), which is power-modulated by a vector network analyzer (VNA), is used to excite the membrane and optothermally drive it into motion. Displacements are detected by focusing a red He-Ne laser beam ($\lambda$ = 632 nm) on the cavity formed by the membrane and Si substrate. The reflected light, which is modulated by the position-dependent membrane motion, is recorded by a photodiode and processed by a phase-sensitive VNA. All measurements are performed at incident laser powers of $P_{red} < 10 \, \mu W$ and $P_{blue} < 0.6 \, \mu W$. It is checked for all membranes that the resonance frequency changes due to laser heating are insignificant. Laser spot size is on the order of ~1 $\mu$m. The uncertainty in measured transition temperatures is estimated from determining the peak position in $-\frac{d(f_0^2(T))}{dT}$ within 2% accuracy in the measured maximum. Information about the reproducibility of measurements is available in Supplementary Note 8.

**Atomic force microscopy.** AFM inspections to determine sample thickness are performed in tapping mode on a Bruker Dimension FastScan AFM. We use cantilevers with spring constants of $k = 30$–$40 \, N \, m^{-1}$. Error bars on reported thickness values are determined by measuring three to five profile scans of the same flake.

## Data availability

The data that support the findings of this study are available from the corresponding authors upon request.

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

## Acknowledgements

M.Š., M.L., H.S.J.v.d.Z. and P.G.S. acknowledge funding from the European Union's Horizon 2020 research and innovation program under grant agreement number 785219 and 881603. H.S.J.v.d.Z., E.C. and S.M.-V. thank COST Action MOLSPIN CA15128; E.C. and S.M.-V. thank ERC AdG Mol-2D 788222, the Spanish MINECO (Project MAT2017-89993-R co-financed by FEDER and the Unit of Excellence 'Maria de Maeztu' MDM-2015-0538) and the Generalitat Valenciana (Prometeo Programme).

## Author contributions

M.Š., M.L., E.C., H.S.J.v.d.Z. and P.G.S. conceived the experiments. M.Š. performed the laser interferometry measurements. M.L. fabricated and inspected the samples. S.M.-V.

and E.C. synthesized and characterized the FePS$_3$, MnPS$_3$, NiPS$_3$, and 2H-TaS$_2$ crystals. M.Š., Y.M.B., and P.G.S. analyzed and modeled the experimental data. H.S.J.v.d.Z. and P.G.S. supervised the project. The paper was jointly written by all authors with a main contribution from M.Š. All authors discussed the results and commented on the paper.

## Competing interests

The authors declare no competing interests.
