## [Peer Review File · Nature Communications]

Reviewers' comments:

Reviewer #1 (Remarks to the Author):

The manuscript by Siskins et al. uses 2D-material-based NEMS devices to study phase transitions in correlated 2D materials. In particular, the antiferromagnet-to-paramagnet transition in FePS₃ (as well as other transitions in other materials) are analyzed. Many results are reported but the most interesting ones, from my perspective, are: i) that it is possible to observe a magnetic phase transition by mechanical means, ii) that the Neel temperature is strain-dependent, and iii) that the dissipation is linked to the phase transition. I am truly impressed by this work: this is a new technique (all-mechanical way to look at phase transitions by linking the thermal expansion to the specific heat), new materials (2D antiferromagnets+other materials), new and interesting results (see above). In fact, it seems that the results from two or three papers are packed into a single manuscript. The paper is very well written (well, except for the abstract in my view), figures are aesthetically pleasing, and the conclusions are original.

I strongly recommend acceptance of this work as-is. The results are motivating and should open a new field of research.

I only have a few questions/comments:

- 1) I feel that the abstract is vague and does not reflect the excitement of the results. I would focus on the main results of the FePS₃ sample.
- 2) Why is the position of the quality factor minimum in Fig. 2d is different from the phase transition point?
- 3) It was not clear how the Neel temperature in Fig. 3d is exactly extracted from the peak in Fig. 3c.
- 4) It was not immediately clear how many samples were measured and how reproducible the results are.

Reviewer #2 (Remarks to the Author):

Review on "Magnetic and electronic phase transitions probed by nanomechanical resonators"

The authors present a new approach to study magnetic and electronic phase transitions in 2D materials. The mechanical detection of these phases relies on associated changes in entropy, and the thermal expansion coefficient, to shift the resonance frequency of the sample. The authors demonstrate success of this approach by detecting the antiferromagnetic phases of FePS₃, NiPS₃, and MnPS₃, as well as the electronic order in TaS₂. New approaches to identify exotic phases that emerge in 2D systems is of general interest because their small samples sizes typically mean that only optical measurements are an option, and thermodynamic measurements are not.

In the case of the materials presented, I think the authors provide compelling evidence that they do indeed detect the discussed magnetic and electronic phase transitions. My biggest concern is that this is not a quantitative approach, in the same sense as specific heat measurements, which can quantify the entropy change associated with such transitions. Indeed, several parameters that rely on estimates from bulk measurements are needed from the literature in order to convince oneself that what's being measured is related to the specific heat. Some prior knowledge is also already known about the type of transition that these example systems undergo. This is fine for demonstrating that a new technique works, but in general this information won't be known so it's unclear how useful this approach will be. One may not be able to deconvolve what's responsible for the transition (changes in

the specific heat or the elastic moduli) or even what's being detected?

Related to this topic, I'm wondering if this approach is very sensitive to thermal conductivity. How does heat from the laser leave the membrane? It's mentioned in the methods section, that this is insignificant, but if one looks closely at figures 2D and 4D, the transition (at even such a large temperature range) shows a slight mismatch between $1/Q$ and the frequency shift, suggesting that these measurements are not purely heat capacity, but rather involve dissipative processes such as thermal conductivity. All thermodynamic transitions should occur at the same temperature, implying that some non-thermodynamic (transport) processes are at play.

It's very clear that there is a big drop in the Q-factor upon cooling near the phase transition. How can we be sure that the change in Q-factor is not causing the observed frequency shift? Because these measurements are done at finite frequency, how do we determine that the change in frequency across the transition is not heavily influenced by dynamics? Both the frequency shift and Q are related to each other via causality and cannot be fully disentangled.

I would be happy to recommend this work for publication if the authors can address these concerns.

Reviewer #3 (Remarks to the Author):

This manuscript describes a new technique for the study of magnetic and electronic phase transitions in layered quantum materials. The method entails the characterization of a drumhead resonator using laser interferometry. The drumhead membrane is the material to be characterized. The authors show that the mechanical cavity resonance changes when going through a magnetic or electronic phase transition in a manner they can connect to the temperature dependent specific heat. They also demonstrate additional behaviors such as strain tuning of the Nernst temperature. The technique is clever and is broadly applicable as a non-contact method to measure phase transitions in novel layered materials. The authors are diligent in their theoretical description of the technique and provide additional examples of successful characterization in the supplement. It is well written, of general interest, and potentially transformative for future investigations of layered quantum materials. I therefore recommend it for publication in Nature Communications without further modifications.

We thank the reviewers for their time and their detailed positive evaluation of our work. We have added additional information on experiments and analysis, which addresses the questions and suggestions of the reviewers and further support the findings reported in the main manuscript. With this letter, we submit the revised manuscript and provide our replies to the questions and comments of the reviewers:

Reviewer #1 (Remarks to the Author):

The manuscript by Siskins et al. uses 2D-material-based NEMS devices to study phase transitions in correlated 2D materials. In particular, the antiferromagnet-to-paramagnet transition in FePS₃ (as well as other transitions in other materials) are analyzed. Many results are reported but the most interesting ones, from my perspective, are: i) that it is possible to observe a magnetic phase transition by mechanical means, ii) that the Neel temperature is strain-dependent, and iii) that the dissipation is linked to the phase transition. I am truly impressed by this work: this is a new technique (all-mechanical way to look at phase transitions by linking the thermal expansion to the specific heat), new materials (2D antiferromagnets+other materials), new and interesting results (see above). In fact, it seems that the results from two or three papers are packed into a single manuscript. The paper is very well written (well, except for the abstract in my view), figures are aesthetically pleasing, and the conclusions are original.

I strongly recommend acceptance of this work as-is. The results are motivating and should open a new field of research.

We thank the reviewer #1 for highlighting the importance of presented results, and for positive feedback on the content and message of the manuscript. We carefully went through reviewer's remarks and provide our answers below.

I only have a few questions/comments:

1) I feel that the abstract is vague and does not reflect the excitement of the results. I would focus on the main results of the FePS₃ sample.

We thank the reviewer for expressing the concerns on the wording of the abstract. In the abstract, we would like to highlight the general nature of the presented methodology. However, we agree with the reviewer on the matter of focus on FePS₃ and thus revised the abstract adding more specific information about the presented findings in FePS₃.

2) Why is the position of the quality factor minimum in Fig. 2d is different from the phase transition point?

The Q-factor minimum, and thus the Q^{-1} maximum in Fig. 2d of the submitted manuscript is off by a single measurement point as compared to the maximum in $c_v \times T$ or $\frac{d(f_0^2)}{dT}$ terms, so the offset is related to the experimental acquisition noise.

In Figure R1, the Q^{-1} terms are displayed and compared to $c_v \times T$ terms for the FePS₃ sample from Fig. 2 (Drum 1) together with the data from an additional sample (Drum 2). We choose the sample parameters (i.e., thickness, t , and drum diameter, d) to be similar for a fair comparison. We note that the position of the peak for Q^{-1} and $c_v \times T$ (and thus $\frac{d(f_0^2)}{dT}$) of the drum 2 corresponds to each other within the measurement resolution.

Figure R1. In both panels: Filled orange dots - measured mechanical damping Q^{-1} as a function temperature. Filled blue dots - normalized $c_v \times T$ term.

We further show reproducibility of this data for drum 2 for three temperature cycles in Fig. R3. We also added this additional data on samples of suspended FePS₃ to Supplementary Section 8 together with a corresponding note in the Methods section of the main text.

3) It was not clear how the Neel temperature in Fig. 3d is exactly extracted from the peak in Fig. 3c.

We extract T_N from the position of the peak's maximum [1]: we take the data point that has the highest value, $\max \left[-\frac{d(f_0^2(T))}{dT} \right]$. To determine the uncertainty in the value of the Neel temperature, as we state in the Methods section «Laser interferometry» subsection of the submitted manuscript, “The uncertainty in measured transition temperatures is estimated from determining the peak position in $-\frac{d(f_0^2(T))}{dT}$ within 2% accuracy in the measured maximum.” In figure R2 below, we visually show the principle. We added an additional note referring to the Methods section in the caption of figure 3 of the main text to make this clearer for the reader.

Figure R2. Example of extraction of the peak position using the maximum in $-\frac{d(f_0^2(T))}{dT}$ from data ($V_g^{\text{DC}} = 0$ V) in Fig. 3c of the main text.

4) It was not immediately clear how many samples were measured and how reproducible the results are.

We thank the reviewer for noting on this ambiguity. In the submitted manuscript and supplementary information, we present data on 2 FePS₃ drums, 1 2H-TaS₂ drum, 1 NiPS₃ and 1 MnPS₃. We also added data on one additional FePS₃ drum to Supplementary Section 8 in a result of preparing this response letter. We added a note on that matter in the Methods section. Apart from the data we report in the submitted manuscript, we measured a total number of more than 10 different drums of FePS₃, NiPS₃, MnPS₃, 2H-TaS₂ and other layered compounds – all showing signatures of a phase transition.

The results for FePS₃ are well reproducible within a single sample with no hysteresis observed in both frequency and Q-factor for multiple temperature sweeps, as shown in Fig. R3.

Figure R3. Multiple temperature sweeps for drum 2 from Fig. R1 and R4.

As displayed in Fig. R4, we compared the T_N detection between two different drums to assess the reproducibility between different devices. The features related to the phase transition are present in both Q factor and frequency in both cases. The small difference in T_N can be attributed to strain variations.

Figure R4. Comparison between two FePS₃ drums.

We also add this additional data to Supplementary information 8 and a note to the Methods section of the main text to make the reproducibility matter clearer to a reader.

Reviewer #2 (Remarks to the Author):

Review on “Magnetic and electronic phase transitions probed by nanomechanical resonators”

The authors present a new approach to study magnetic and electronic phase transitions in 2D materials. The mechanical detection of these phases relies on associated changes in entropy, and the thermal expansion coefficient, to shift the resonance frequency of the sample. The authors demonstrate success of this approach by detecting the antiferromagnetic phases of FePS₃, NiPS₃, and MnPS₃, as well as the electronic order in TaS₂. New approaches to identify exotic phases that emerge in 2D systems is of general interest because their small samples sizes typically mean that only optical measurements are an option, and thermodynamic measurements are not.

We thank the reviewer for highlighting the results our work, positive feedback on the message of the manuscript and for evaluating the perspectives of the use for the presented approach for identification of new exotic phases in 2D systems. We carefully address his/her indeed interesting questions below.

In the case of the materials presented, I think the authors provide compelling evidence that they do indeed detect the discussed magnetic and electronic phase transitions. My biggest concern is that this is not a quantitative approach, in the same sense as specific heat measurements, which can quantify the entropy change associated with such transitions. Indeed, several parameters that rely on estimates from bulk measurements are needed from the literature in order to convince oneself that what’s being measured is related to the specific heat. Some prior knowledge is also already known about the type of transition that these example systems undergo. This is fine for demonstrating that a new technique works, but in general this information won’t be known so it’s unclear how useful this approach will be. One may not be able to deconvolve what’s responsible for the transition (changes in the specific heat or the elastic moduli) or even what’s being detected?

We agree with the referee on this point. A quantitative analysis of the specific heat measurement with this method is only possible if several material parameters (Young's modulus, Poisson ratio, etc.) are known, and are not changing significantly at the phase transition. Literature values (experimental or theoretical) of the relative change of these parameters from bulk samples can give an indication of their potential effects. Under the condition that these are known and the assumptions hold, the method can provide a reasonable quantitative estimate of c_v , as demonstrated in Fig. 4b of the submitted manuscript and in Section 2 of the Supplementary information. This is also discussed in the main text of the submitted manuscript in "Resonance and specific heat" and "Detecting phase transitions" subsections and "Discussion" section. In the scenario when, for example, changes in Young's modulus as a function of temperature do play a significant role, additional measurements of the Young's modulus (e.g. by AFM or nonlinear dynamics [2]) are needed for a quantitative analysis of the specific heat with the presented method. Also, methods to extract the required thermal parameters as a function of temperature are available [3],[4]. We note that although it is more involved, the different analyses could still be done on the same device using a very similar experimental organization. Once all parameters have been characterized as a function of temperature, combining them will allow deconvolving their relative contributions to the phase transition (specific heat versus other contributions).

Moreover, we note that, even with only a qualitative detection of a phase transition by considering just the anomaly in the $-\frac{d(f_0^2(T))}{dT}$ term, the method is still very useful because (i) it allows a non-contact detection of a phase change in systems that are hard to study otherwise (e.g. ultrathin insulating antiferromagnets), and (ii) it is sensitive to multiple types of phase transitions. Using only qualitative detection, the method can be broadly applicable to study the strain dependence of the transition temperature in 2D layered materials, as demonstrated in Fig. 3 of the main text, as well as the layer number dependence for systems with a strong interlayer coupling (e.g. NiPS₃ [5]).

Related to this topic, I'm wondering if this approach is very sensitive to thermal conductivity. How does heat from the laser leave the membrane? It's mentioned in the methods section, that this is insignificant, but if one looks closely at figures 2D and 4D, the transition (at even such a large temperature range) shows a slight mismatch between 1/Q and the frequency shift, suggesting that these measurements are not purely heat capacity, but rather involve dissipative processes such as thermal conductivity. All thermodynamic transitions should occur at the same temperature, implying that some non-thermodynamic (transport) processes are at play.

We agree with the reviewer #2 that it is indeed important to ensure that the heat from the laser does not contribute significantly to the average temperature of the membrane. However, if thermal conductivity plays a role then there should be a laser power dependence on the phase transition temperature.

To ensure that this is not the case, we verify by laser-power dependent measurements, that the laser power of both blue and red laser is small enough to not affect the resonance frequency and the Q factor of the membrane. This is a standard procedure for all measurements presented in the submitted manuscripts, as also mentioned in the Methods «Laser interferometry» subsection. This condition ensures that laser heating is insignificant.

Considering figures 2d and 4d and the corresponding Q^{-1} trend: as already mentioned above for the case of Fig. 2d, maxima in Q^{-1} in both figures are off by only 1-2 measurement points as compared to the maximum in the $c_v \times T$ or $\frac{d(f_0^2)}{dT}$ terms, which is related to the experimental noise in the data. Data shown in Fig. R1, R3 and R4 in response to reviewer #1 comment supports this argument. We also added this additional data to Supplementary Section 8 and included a note in the Methods section of the main text.

It's very clear that there is a big drop in the Q-factor upon cooling near the phase transition. How can we be sure that the change in Q-factor is not causing the observed frequency shift? Because these measurements are done at finite frequency, how do we determine that the change in frequency across the transition is not heavily influenced by dynamics? Both the frequency shift and Q are related to each other via causality and cannot be fully disentangled.

We thank the reviewer for expressing his/her thoughts on this interesting scenario. For an underdamped harmonic oscillator, $f = f_0 \sqrt{1 - \frac{1}{4Q^2}}$. Since the Q-factor for the sample presented in Fig. 2 drops from 5900 to 480 next to the phase transition, we expect the effect on f to be of the order of kHz, which is insignificant compared to the ~ 2 MHz change in frequency, as shown in the figure 2a of the main text. We add a clarifying note on this matter to the Supplementary Information section 2.

I would be happy to recommend this work for publication if the authors can address these concerns.

We thank the reviewer for his/her time and recommendation.

Reviewer #3 (Remarks to the Author):

This manuscript describes a new technique for the study of magnetic and electronic phase transitions in layered quantum materials. The method entails the characterization of a drumhead resonator using laser interferometry. The drumhead membrane is the material to be characterized. The authors show that the mechanical cavity resonance changes when going through a magnetic or electronic phase transition in a manner they can connect to the temperature dependent specific heat. They also demonstrate additional behaviors such as strain tuning of the Nernst temperature. The technique is clever and is broadly applicable as a non-contact method to measure phase transitions in novel layered materials. The authors are diligent in their theoretical description of the technique and provide additional examples of successful characterization in the supplement. It is well written, of general interest, and potentially transformative for future investigations of layered quantum materials. I therefore recommend it for publication in Nature Communications without further modifications.

We thank the reviewer for the positive assessment of our work and for stressing the applicability of the demonstrated detection methodology and potential of the use in further investigations of quantum materials.

References:

- [1] Landau, L. D., Pitaevskii, L. P. & Lifshitz, E. M. *Electrodynamics of continuous media*, vol. 8 (Butterworth, New York, 1984), 2 edn
- [2] Davidovikj, D., Alijani, F., Cartamil-Bueno, S.J. et al. Nonlinear dynamic characterization of two-dimensional materials. *Nat Commun* **8**, 1253 (2017).
- [3] Morell, N. et al. Optomechanical measurement of thermal transport in two-dimensional MoSe₂ lattices. *Nano Lett.* **19**, 3143–3150 (2019).
- [4] Dolleman, R. J. et al. Transient thermal characterization of suspended monolayer MoS₂. *Phys. Rev. Mater.* **2**, 114008 (2018).
- [5] Kim, K., Lim, S.Y., Lee, J. et al. Suppression of magnetic ordering in XXZ-type antiferromagnetic monolayer NiPS₃. *Nat Commun.* **10**, 345 (2019).

REVIEWERS' COMMENTS:

Reviewer #2 (Remarks to the Author):

The authors adequately addressed my concerns. Thank you for the detailed responses, and updates to the manuscript. I recommend this paper for publication.